# The Meaning of the Common World in Perioperative Nursing Care; A Hermeneutic Study

**Susan Lindberg [1,2,*] and Gudrun Rudolfsson [3,4]**

1   Department of Anaesthesia and Surgery, Skaraborg Hospital, SE-54185 Skövde, Sweden
2   Department of Research and Development, SkaS, SE-54185 Skövde, Sweden
3   Department of Health Sciences, University West, SE-461 86 Trollhättan, Sweden
4   Faculty of Nursing and Health Sciences, Nord University, 8049 Bodø, Norway
*   Correspondence: susan.lindberg@vgregion.se

**Abstract:** The aim of this study is to bring forth the meaning of the common world as it appears in perioperative nursing care. We employed the epistemological standpoints of preunderstanding, the hermeneutic spiral and fusion of horizons grounded in Gadamer's hermeneutic philosophy as well as Eriksson's Theory of Caritative Caring based on the ontology of caring science, where caritas is the basic motive and ethos of caring. Four hermeneutic spiral activities were performed, consisting of a mimetic presentation bearing the ontological depth of the common world, its distinctive features, the universal and lasting and finally, the truth inherent in the common world. The inherent truth of the common world is the prevalence of harmony, wholeness and the idea of love, mercy and reverence for human dignity. The common world brings ethics to existence, achieved by the word of honour, which in its true being makes visible the universal and ontological horizons of a common reality. The common world is the creation of a hermeneutic movement inside each suffering human being, where the boundless life-giving time represents the inhabited movement of time, like coming home.

**Keywords:** common world; distinctive features; Gadamer; hermeneutics; perioperative

## 1. Introduction

This hermeneutic study is a continuation of previous qualitative research dealing with perioperative dialogues conducted between adult patients, children with special needs, and perioperative nurses, i.e., nurse anaesthetists or operating room nurses. The perioperative dialogue is 'Those nursing actions and activities performed by the nurse anaesthetist or operating room nurse in the pre-, intra-, and postoperative phases of the patient's surgical procedure' (von Post 1999, p. 73). The word 'peri' refers to time in the sense of closest to the time the patient shares with a perioperative nurse (Lindwall and Iréne 2009), a sharing that presents the patient with an opportunity to be a part of a common world (Rudolfsson 2007). The common world is tentatively defined as 'a means for patient and nurse to become present for the other, to be in communion, face each other in a process of a continuous whole and move towards health, characterized by a mutual expression of acceptance, where each person brings him or herself into the process, where acceptance means conducting oneself in an authentic way' (Rudolfsson 2010, p. 33).

According to Eriksson (Lindström et al. 2006), human being is fundamentally dependent on communion; she/he is dependent on another and it is in this common world that the core of all humanity rests. Gadamer [1975] (1989) also mentions the common world as a path to humanity with the other, taking place within the language and a part of the universal dimension of life. Furthermore, Arendt [1971] (1981) underlines that if we lose our sense of a common world or are forcibly kept from sharing it with our fellow human beings, we lose something of our humanity. However, the common

world has not been widely researched. Gadamer [1975] (1989) holds that one reason may be that the common world is one of the most difficult phenomena to grasp, not because it is too remote but because it is so close to us. We therefore believe it is important to understand what the common world means from the perspective of the patient in the context of perioperative nursing care.

## 2. Aim

The aim of this study is to bring forth the meaning of the common world as it appears in perioperative nursing care.

## 3. The Perioperative Dialogue

The perioperative dialogue is an ideal model on which is based the perioperative nursing care due to its humanistic and caring science perspective (Eriksson 2002). The purpose is to protect human dignity, alleviate suffering and create wellbeing through the nurse's continuity of care (von Post 1999). Using dialogue as a point of departure, the perioperative model is based on Martin Buber´s philosophy of dialogue as inter-human meetings where two people face each other (Buber 1990).

In the pre-operative dialogue, the patient and nurse meet before surgery. The aim of this dialogue is to clarify issues that need to be explained to the patient and to plan the intra-operative nursing care. The intra-operative dialogue begins when the patient meets the same nurse in the Operating Department. The nurse again explains what will happen and cares for the patient on the basis of the agreed caring plan. In the post-operative dialogue, the nurse returns to the patient to enquire how she/he feels and to provide her/him with a chance to ask questions about the operation. In Sweden, parental presence is mandatory during the perioperative process. Accordingly, children and parents are not separated until the child is asleep.

In a literature review, Lindwall and Iréne (2009) concluded that the value of the continuity created by the perioperative dialogue generated conditions for the nurse to care for the patient in a dignified way.

As a consequence, the perioperative dialogue sets the prerequisites for the common world because it allows the patient and nurse to be in a communion in which new ways of thinking and acting may emerge (Lindberg 2013; Lindwall and Iréne 2009; Rudolfsson et al. 2003b; Rudolfsson 2007). With this study, we, as nurse researchers, wish to take responsibility for our previous empirical and theory-based research findings on the perioperative dialogue by interpreting and translating them in order to present the common world of patients and nurses.

## 4. Empirical Underpinning of the Study

The empirical research findings are derived from seven peer reviewed articles (Lindberg and Post 2005; Lindberg and Post 2006, Lindberg et al. 2012; Lindberg et al. 2013; Rudolfsson et al. 2003a; Rudolfsson et al. 2007a, 2007b). All seven studies were qualitative. One had a grounded theory approach and six a hermeneutic approach. In addition, one employed a secondary analysis, one a synthesis and one the critical incident technique. Data were collected by means of semi-structured and conversational interviews, semi-structured interviews combined with direct observations, play interactions inspired by Gadamer (1998) and written narratives.

The articles contain data from 90 patients aged between five and 76 years, in addition to 12 biological or adoptive parents, seven mothers and five fathers, whose children were diagnosed as having a severe autistic disorder. The perioperative dialogues were conducted by 25 nurses (15 nurse anaesthetists and ten operating room nurses) with ten to 34 years of perioperative practice experience. These articles form the basis of the two authors' respective research in which theory and empirics are interpreted in the light of each other (Lindberg 2013; Rudolfsson 2007).

The aim and the main findings of the seven empirical studies are presented below (Table 1).

**Table 1.** Aims and main findings of the empirical studies.

Study I

The aim was to describe patients' experiences of the perioperative dialogue. 'Making time for me' seemed to be the main pattern related to the patients' experiences of the perioperative dialogue. The time with the nurse comprised 'comforting me' and 'becoming involved'. The patients felt involved, relieved and at ease when they realized that the nurse will be there to support them throughout the surgery (Rudolfsson et al. 2003a).

Study II

The aim was to gain a new understanding of what constitutes caring in the perioperative dialogue from the perspectives of the patients, nurse anaesthetists and operating-room nurses. 'Caring as a solemn vow or promise' emerged as a new horizon of understanding from both the patient and the nurse perspective. 'Vow' was the main theme that captured different meanings of caring in the perioperative dialogue. When the nurse anaesthetist or operating-room nurse promised the patient that he will be allowed to be himself, to safeguard his welfare, to guide him through the surgery and take responsibility for honouring her promise to the patient, caring became visible to the patient as meaning that the nurse takes him seriously, creates a calm atmosphere, thus implying an assurance that he can hand over responsibility to the nurse (Rudolfsson et al. 2007a).

Study III

The aim was to synthesize the findings of two previous qualitative studies to generate a new understanding of and consensus on health in the perioperative dialogue from the perspectives of the patients, nurse anaesthetists and operating-room nurses. The findings indicate that health in the perioperative dialogue means being in communion in a continuous whole of care. In a 'continuous whole', the patient feels that he is a unique human being, as the nurse treats him as an active partner and a valuable source of information. As health is the ultimate goal of caring, it becomes visible when the nurse offers time to the patient, thus leading to conditions for communion. Health becomes visible as a source of strength, as the perioperative dialogue can help to improve the nurse's sense of wellbeing when she is able to 'walk together' with the patient and successfully create a caring relationship (Rudolfsson et al. 2007b).

Study IV

The aim was to describe experiences of the perioperative dialogue of children with special needs. In the course of the three phases of the perioperative dialogue it was possible to create room for new opportunities through the children feeling invited to be participants in their own care. Several children described their participation in the dialogue as getting to know the nurse and that a sense of togetherness is created based on the nurse anaesthetist's continuous presence. In the course of being prepared for and then being administered the anaesthetic by the same nurse the necessary knowledge becomes available to both the nurse and the child with special needs alike, thus which at the beginning may seem impossible actually turns out to be easy (Lindberg and Post 2005).

Study V

The aim was to describe what may help children to manage their fear of general anaesthesia during the perioperative dialogue with the nurse anaesthetist. How do children describe what it feels like to be a part of this care planning process? The result revealed that confidence emerged as an answer to the research question. The way the children move from fear to confidence was described as: *To be given time, to be taken seriously, to participate in decisions, to be able to help, to know that it will go well and to enjoy the return to the hospital.* Being a part of the perioperative dialogue made room for a caring communion in which children who are unlikely to benefit from routine interventions designed to reduce perioperative anxiety can confidently surrender themselves and their body into the hands of the nurse anaesthetist (Lindberg and Post 2006).

Study VI

The aim was to obtain an understanding of what parents of children with severe autism experience in connection with their child's anaesthesia in the presence and in the absence of the perioperative dialogue. The findings revealed that the parents experienced previous anaesthetic care of their child in the absence of the perioperative dialogue as the suffering of care, while the perioperative dialogue alleviated the suffering of care because of the continuity provided by the nurse anaesthetist. The path through the suffering of care emerged as: *A hopeless struggle, unspeakable suffering and a disgraceful scenario.* Children and parents were subjected to mute suffering in a double sense and their dignity was violated. The perioperative dialogue created a place where the movements of play become visible, comprising: *Being received by warm hands, being received by a familiar face and a subtle interplay.* The children and parents acquired new horizons, experienced as the alleviation of suffering (Lindberg et al. 2012).

Study VII

The aim was to explore, exemplify and discuss how a participatory hermeneutic method designed for children with special needs can be developed from the concept of play in a caring context. The finding shows that Gadamer's 'give-and-take game' evokes the ethics of play, the practice of the finest art which takes the child and her/his parent to the play of truth. Play, as both hermeneutic interpretation and the substance of caring, can thus be directed toward the child in a perioperative caring context, not by narrowing methods but through the art of reading and interpreting the child as a secret script, even if only mere glimpses of it appear (Lindberg et al. 2013).

## 5. Ontological Standpoints

The ontological issues associated with the core of caring in the common world are of primary importance for this study, which is guided by Eriksson's theory of caritative caring (Lindström et al. 2006). The theoretical perspective is based on the ontology of caring science, where caritas is the basic motive of caring and ethos the ultimate meaning of the caring context. The common world means being part of the same wholeness and implies an invitation from two parties who both open up and that the patient and nurse have something in common, i.e., they are in a communion. The attempt to find order in the common world of caring means trying to identify a structure based on the caritative caring theory and a human science perspective, which at the same time encompasses the researcher's preunderstanding. The patient, the suffering human being, is regarded as an entity consisting of body, soul and spirit. The suffering human being is the focus of perioperative caring, where the care provided should be based on love and mercy, built on sympathy and compassion (Eriksson 2002; 2006; Lindström et al. 2006). Eriksson (2002) emphasizes that in a symbolic sense, one's own ethos calls for seeing the human being, the patient, as a secret script that each person must learn to read and interpret.

## 6. Epistemological Standpoints

The epistemological standpoints of this study have their roots in the philosophy of Gadamer (1976, [1975] 1989), where the emphasis is on interpreting and understanding the human world. This understanding emerges through dialogue because it is only through dialogue that human beings can give meaning to the world. In addition, Gadamer claims that language 'is only fully what it can be when it takes place in dialogue' (Gadamer 1976, pp. 127–28). Language is the human being's unique way to have a world and furthermore, 'it constitutes the common world in which we live, and to which belongs also the great chain of tradition and historical horizon' (Gadamer 1976, p. 65). If we compare this to real life, this we-ness involves that the other, whether a human being or a written text, appears as a partner, co-creator and another who is both different yet close enough to be understood, taken seriously and acknowledged. In doing so we can escape the vicious circle that obstructs the understanding and enter into the dialectic movements between the whole and the parts, termed by Gadamer as a spiral activity in which the interpreter of the text is drawn into new circles of the unexpressed or unsaid, which continue to pose new questions that prompt her/him to search for answers (Gadamer [1975] 1989).

What remains to be added is that no text is final and therefore the authors of this study are called to make the text from their previous research speak again, to ensure that it is heard. Gadamer [1975] (1989) specifies this by stating that a new understanding may emerge when the interpreter highlights some features from the original text that are central to the phenomena under study. As Gadamer [1975] (1989) states, this means that when a text is truly interpreted it does not become a copy but instead a radical transformation because it bears ontological depth in relation to human reality. A radical transformation reveals the truth in that reality (Gadamer [1975] 1989).

## 7. Method

Gadamer believes that the understanding of the common world is always hermeneutic (Gadamer 1976). Consequently, the philosophical hermeneutics of Gadamer [1975] (1989) was used as a method for interpreting the text. This meant searching for deep knowledge aimed at opening up and expanding the issue, as well as acquiring a sensibility in relation to the research process. Key elements in Gadamer's hermeneutics are preunderstanding, the hermeneutic spiral and fusion of horizons. Together, these three elements create the dialectical movements, which are at the heart of hermeneutic text interpretation (Gadamer [1975] 1989). The fact that the text came from the authors' own research and that both authors have decades of perioperative nursing care experience means that understanding can be directed towards an opening up of new horizons. However, this can also be a hindrance if the researcher is bound by her/his preunderstanding and therefore unable to open up by moving further. During

the interpretation an open and dialectical approach enabled receptiveness to let the text speak about unfamiliar meanings, the otherness that waited to be incorporated in the authors' horizons as well as in the horizon of the text (Gadamer [1975] 1989). The interpretation of the text was made on the basis of four spiral activities of interpretation and abstraction as described by Gadamer [1975] (1989).

To ensure that the audience, i.e., the potential reader can participate in the bringing forth of the common world as described by Gadamer, it was found suitable to weave together the hermeneutic text interpretation with the gradually emerging understanding (Gadamer [1975] 1989). In the following, the term nurse refers to both nurse anaesthetists and operating room nurses, while to enhance the flow of the text the nurse is denoted 'she' and the patient 'he'.

## 8. Mimetic Presentation of the Common World—The First Spiral Activity

Gadamer [1975] (1989) describes mimesis as an interpretative activity in which the interpreter highlights some extremely important elements from the text by omitting, uncovering, fixating and bringing forth the essentials and at the same time concealing other elements.

The findings from previous research were initially read with an open mind, which means that the authors asked what the text has to say. While reading, questions emerged such as: Are there descriptions of the common world in the text? The text answered: Yes, there are. Significant expressions and metaphors that emerged from the text were then formulated into meaning by comparing the whole with the parts so that the substance of or the content in these can be uncovered. In order to overcome arbitrariness, this spiral activity was centrifugally expanded (Gadamer [1975] 1989) by using philosophical as well as caring science literature thereby reshaping the reality into a mimetic presentation of the common world. The mimetic presentation is summarized in four abstracted themes comprising a common language. More specifically, the common language found in the themes comprises different yet linguistically unified expressions, the content of which did not differ between children and adults. Gadamer [1975] (1989) describes this as the universal horizon of hermeneutic experience.

### 8.1. 'There You Are'—The Accessibility of the Face

In the common world the patient and nurse face each other in a process of a continuous whole (Rudolfsson 2010), which distinguishes itself as the various but nevertheless unifying nuances of saying 'there you are'. Having access to a familiar face, a concrete human being with a name can be perceived as a thread of meaning. The face talks to the receiver, it expresses continuous messages from a 'You' to an 'I', where through the universe of language, the common language speaks and can be heard (Gadamer [1975] 1989). Through the common language the world becomes a common world that unites the individual and the universal. Moreover, it is a world in and through which the accessibility of the face expands the space in the direction of infinity and boundlessness (Gadamer [1975] 1989, 1998; Levinas 1969).

### 8.2. Being in a Harmonious Atmosphere—To Be at Home with Oneself

In the common world a harmonious whole is created, achieved by the nurse's giving of her whole self. Her voice and hands bear the message of hospitality that includes the beautiful and the good (Gadamer [1975] 1989), which in the common world gives the patient a perceptible sensation of being in a harmonious atmosphere. In the harmonious atmosphere the nurse's head and heart rule her hands, which Gadamer (1998) termed the intelligent hands that manifest a sharing with one another. Being at home with oneself within the peaceful communion in the common world may give the human being a sense of freedom and a moment of respite from suffering (Gadamer [1975] 1989, 1998; Levinas 1969).

### 8.3. 'There You Are'—As You Promised

The common world enable the creation of a world where mutual trust can grow and function in an undisturbed way, achieved by means of the truly spoken vow, the nurse's promise to remain by the

patient's side. Together they shape a harmonious entirety in which the word of the promise reaches its true meaning as the gift of faith (Gadamer 1998). Through the experience 'There you are'—as you promised, the world transforms into a common world of humanity and faithfulness that in the words of Gadamer distinguishes itself as true, beautiful and good (Gadamer 1998).

*8.4. Being in the Life-Giving Time—Coming Home*

In the common world a two-fold time perspective becomes evident that distinguishes between an outer and an inner time dimension (Nurminen 2009). The experience of the outer time dimension can be attributed to everyday life, where life takes its habitual course. The inner time dimension can be perceived as a settlement—a home, where the movement of time is inhabited or standing still. This inner time dimension is life-giving and perceived as both fragile and precious. Getting and giving time constitutes a movement towards the other person, which for the patient means not only perceptions of being at home with himself but also the powerful sensation of coming home. In the view of Levinas, the time and the homeliness shared with the other are endless and give the human being the sensation of an inherent opportunity beyond the boundaries of space and time (Colin 1990). Space that allows the word to run its full course cannot be hasty but requires another time than that governed by efficiency (Martinsen 2002). The heart of the life-giving time in the common world rests in togetherness, where through ethical mercy the suffering human being is given priority (Levinas 1969). This time perspective must be considered pivotal, not only for keeping the unity of patient and nurse together as a whole, but also for their sharing of the same wholeness, the horizons of the past, the present and the future in the common world they have jointly created.

## 9. Distinctive Features of the Common World—The Second Spiral Activity

The second spiral activity comprised finding the questions asked by the text. During this transcendence of horizons, it became obvious that the text wanted to tell something about the distinctive features that elevate mimetic presentation to the level of ontology (Gadamer [1975] 1989). Consequently, additional theories from caring and human science (Eriksson 2007; Martinsen 2006; Watson 2003) were explored before interacting with the text again, using the chosen theories to provide an understanding on a deeper level. Gadamer [1975] (Gadamer [1975] 1989) describes this as the way to expand the ontological possibilities of human thought.

In the common world, the 'great nest of becoming' (Benner 2016) involves experiences of spiritual rest and calm as well as sensations of rootedness in the patient.

The gift of calmness begets calm and an invitation, an aspect described by Watson as extremely important in the shared human connection (Watson 2003). Eriksson (2007) states that a living communion between the patient and the nurse is necessary for the patient to feel invited as a guest of honour. By saying '*There you are*' the patient showed how the nurse's face invited him into a meaningful world (Levinas 1969), the common world, resulting in the joy of recognition. Levinas further states that to recognize the other is to give in generosity (Colin 1990; Levinas 1969), which in this study became visible when the patient gave something of himself to the nurse by inviting her to share his world. This is an important reminder of the power of the ethics of the face, achieved by the nurse's way of saying 'Here I am' (Levinas 1969). To wholeheartedly make space and time for the patient with reverence for his holiness and dignity is not only the gift of a welcoming presence (Eriksson 2007) but also one of the most important gifts the nurse can give to the patient in his efforts towards an authentic homecoming in the world of caring (Martinsen 2006). Hence, the world becomes open to both of them as a common world.

## 10. The Universal and the Lasting in the Common World—The Third Spiral Activity

In view of Gadamer's statement that the common world is a universal dimension of human life (Gadamer [1975] 1989), it became necessary to put the following question to the text: What is the universal and lasting in the common world? By reading the text repeatedly and actively dwelling on

similarities as well as differences across texts, a new understanding of the content began to emerge. In Gadamer's hermeneutics this type of bringing forth means being edified and seeing the world in a new light, thereby sharpening the sight for truth (Gadamer [1975] 1989).

The common world arises solely due to the fact that one person donates the world to the other. Renouncing one's own world means it is open to both, whereby the faithfulness inherent in the nurse's giving of herself introduces the boundless into the language and endows it with meaning. What is said is a sign of feeling at home in every sense of a vibrant communion, where the language's abode is created in the dialogue about the common world. Together they establish a common reality where the word reaches its completion when the patient understands something of the perioperative world. By its very existence the communion between the patient and nurse seems to grow, leading to them experiencing themselves as richer through the reciprocity that is created, which depends on the participation of both. Accepting a common world means the beginning of a common creation. The spoken word is between the patient and the nurse, which Buber (1990) denotes as the sphere of between. This means that when the word is spoken, it belongs to the sphere between the patient and the nurse. The dialogue between the patient and the nurse implies balancing one's individual freedom with the total gift of oneself. The common world includes a special responsibility on the part of the nurse in order for the patient to feel recognized and accepted in his uniqueness, thus enabling him to place himself in the nurse's hands. The familiar face of the nurse gives the patient a sense of security and a certainty that he can trust the nurse and tell her what he does not tell anyone else. It is in the dialogue that words and stillness spread out and through the word the patient sees his reality in a new light. Caring obtains its deepest meaning through the implementation of the caritas motive, total human love.

## 11. Towards a New Understanding—The Fourth Spiral Activity

In the fourth spiral activity the authors sought to make the common world present in its dynamic fullness (Gadamer [1975] 1989). The authors' task was to give the language a written form so that 'what is' was brought to light, thereby sharpening the sight towards a new horizon. In this new horizon, the new understanding, the ontological, the ethical and the contextual interpretations were interwoven, giving rise to not only a radical transformation of the meaning of the common world but also a transformation into the truth. This occurrence is the connection between what is discerned as beautiful and what is to be understood. Gadamer [1975] (1989) draws attention to an understanding that is not proven or entirely true but must be applied to what stands out from the possible and supposed, as when the light breaks through.

## 12. Fusion of Horizons—The New Understanding

A common world is a world where harmony, wholeness and the idea of love, mercy and reverence for human dignity prevail. This includes a spiritual element that makes the human being universal, which is expressed in love, allowing the entity of the human being, consisting of body, soul and spirit, to attain the horizon of true homecoming. Moreover, the common world brings the foundation of ethics to existence, achieved by the word of honour, which in its true being makes visible the universal and ontological horizons of a common reality (Gadamer [1975] 1989; Levinas 1969). The common world creates a hermeneutic movement inside each suffering human being, where the boundless life-giving time represents the settlement where the movement of time is inhabited, like coming home. In the common world this movement follows a different hermeneutic pattern than that of question and answer, namely gift and invitation. This interpretation was deemed true as Gadamer stresses that hermeneutics is not merely a method but a way of comporting oneself, a mode of being (Gadamer 1976, Gadamer [1975] 1989, 1998).

Based on the new understanding the following definition of the nature and meaning of the common world emerged: The common world comprises a life-giving time where the power of beauty, goodness and truth expands reality in the direction of eternity and boundlessness, making the universal

and ontological standpoints visible. The common world entails the invitation to and welcoming of the other, creating a hermeneutic movement inside the suffering human being expressed as sensitivity, faithfulness and love, allowing the whole human being to attain the horizon of true homecoming.

## 13. Methodological Reflections

No previous research was found that focused on the common world *per se* or its meaning in perioperative nursing care. To discover meanings, dimensions and connections that may remain concealed in purely empirical material, it was deemed that the methodological approach of this paper should preferably be grounded in Gadamer's philosophic hermeneutics (Gadamer 1976, 1989, 1998). The meaning of a text can never be exhausted. The constant revision of a text is how understanding is achieved and how preunderstandings are refined. Being aware of the prejudices ruling our own understanding was crucial so that the text can be isolated and valued in its own right. However, a permanent challenge is that prejudices are intrusive and become an integral part of understanding, something that one should be attentive to and aware of. Adhering to Gadamer [1975] (Gadamer [1975] 1989) meant that the aim of the interpretation was to manifest textual meanings rather than individual ones. This approach was here considered both a philosophy of understanding and a fruitful way of textual interpretation along with bringing forth the meaning of the common world. This implies, as Gadamer [1975] (Gadamer [1975] 1989) puts it, that when something is truly interpreted it reveals not only our shared ontology but creates a new one. It was essential that the meaning of the parts, i.e., the seven empirical studies, should be examined by being related to the text as a separate context from which a whole gradually emerged, thus making it more than the sum of the individual parts. Finally, through the dialogue between the whole and the parts to a new whole, the goal of the hermeneutic spiral was achieved, namely a reasonable interpretation and understanding of the meaning of the common world as it appears in perioperative nursing care.

Truthfulness and transparency were enhanced by describing the various steps in the interpretation as thoroughly as possible and how the new understanding gradually emerged. The differences in the patients' gender, age, way of expressing themselves and the fact that parents' experiences were included, assisted in broadening the horizon of the common world as well as supporting the evidence and enhancing transferability. Gadamer [1975] (Gadamer [1975] 1989) concern was to expand the concept of truth to other forms of truth in such a way that it is not restricted to the limiting and traditional way of viewing truth. Being truthful in the interpretation process meant that we were eager to present the conditions under which the actual statements emerged in the understanding of the truth of the common world.

## 14. Implication and Future Research

It will be desirable to regain the hermeneutic dimension of gift and invitation in the caring context, as this paper demonstrates its importance. The focal element might be nurses listening to the voice of their own heart, involving the dialogue between reason and will, good and evil. The common world 'There you are'—the accessibility of the face, implies the nurses' active presence where they use their energy in a tireless struggle to be present. The common world of patient and nurse is a world that unites the individual and the universal in an uninterrupted context, which may be more necessary than ever in the society today. Further research should deal with the fundamental nature of the gift and true hospitality as well as faithfulness in the world of caring.

**Author Contributions:** Both authors (S.L., G.R.) contributed to all phases of the work on the manuscript; the design, analysis, and interpretations. The authors drafted the manuscript together and both have approved the final version.

**Funding:** This study was funded by the Skaraborg Institute for Research and Development, Sweden.

**Conflicts of Interest:** The authors declare no conflict of interest.

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
