# Peer review of "The Meaning of the Common World in Perioperative Nursing Care; A Hermeneutic Study"

_humanities, doi:10.3390/h8030132_

Round 1
Reviewer 1 Report
This paper has a potential to be improved. There are no basic sources of the perioperative dialouge, an ideal working model within perioperative nursing care. What is the perioperative dialouge? The reader needs more information about earlier research within the model of perioperative dialouge.The two authors have use their dissertation but they have to write about their research instead. Data are from 7 articles. Tabl 1 is incomplete how to understand this?
on page 4 they present the common world is central in caring science without ref..why?
The aim is incomplet to the results of this study. Please make it more clear. There are no method and The authors have used Gadamers philosohy without argument why.The section with ontological standpoint is too extensive. The section of hermeneutic text interpretation need to be rewrite so readers can understand how the interpretation have been done .The result is less from the childrens perspectiv who participated in the perioperative dialouge. What is the common world for them?
What is the new understanding in this study? Describe the substance in the perioperative dialouge?
In the implication the authors need to explain why should nurses and nurse leaders use the perioperative dialouge. What qualities are required by nurses to be in a common world?
Funding NO but Skaraborg institut was funding this reseach. Is this ok?
the references must be reviewed.
Author Response
Authors revision letter, Manuscript ID: humanities-493715
Thank you for your valuable comments to our manuscript, and for given us the opportunity to review and improve the manuscript. We have considered all recommendations and comments from the reviewers, below you find a table with our point by point responses. The changes made in the manuscript are highlighted in bold font.
From reviewer 1: |
Response |
There are no basic sources of the perioperative dialogue, an ideal working model within perioperative nursing care. What is the perioperative dialogue? The reader needs more information about earlier research within the model of perioperative dialogue |
We have added a paragraph titled The perioperative dialogue with this requested information. |
The two authors have used their dissertation but they have to write about their research instead. Data are from 7 articles. Tabl 1 is incomplete how to understand this? |
We agree. Thus, we have added a paragraph titled Empirical underpinning of the study in which our published research from the perioperative dialogue is described including methods for data collection data analysis and research participants. Hopefully it will be clearer that the aims and results in Table 1 are derived from this research. We hope that this introductory paragraph will serve as a complement to Table 1.
|
on page 4 they present the common world is central in caring science without ref..why? |
We have added three references according to the common world in relation to the context of caring science |
The aim is incomplete to the results of this study. Please make it clearer. |
The aim is revised in order to be more complete in relation to the results of this study. |
here are no method and the authors have used Gadamers philosohy without argument why. |
We have omitted the title hermeneutic text interpretation and named the paragraph Method. Under the title Method we have added with arguments why we found Gadamers hermeneutics suitable. |
The section with ontological standpoint is too extensive. |
The section with ontological standpoint has been shortened with 50 words. |
The section of hermeneutic text interpretation needs to be rewrite so readers can understand how the interpretation have been done. |
We have rewritten the hermeneutic text interpretation and it is now integrated in all the four spiral activities following the hermeneutic interpretation movements. |
The result is less from the children’s perspective who participated in the perioperative dialogue. What is the common world for them?
|
We believe that it is no difference between the children and the adults in the experience of the common world, which became obvious in the new understanding, presented under the heading of Fusion of horizon. |
What is the new understanding in this study? |
On pages 15-16 we have divided the text under the heading of “Towards a new understanding” into two parts and added the heading Fusion of horizon-the new understanding in order to make the new understanding explicit. |
Describe the substance in the perioperative dialogue?
|
We are not sure if we understood this comment but in earlier research the substance in the perioperative dialogue has been described as for example continuity of care, play on the stage of caring, promise of care, making time for the patient. |
In the implication the authors need to explain why should nurses and nurse leaders use the perioperative dialogue. What qualities are required by nurses to be in a common world?
|
We agree. The implications have been revised and rewritten in relation to qualities, required by nurses to be in a common world. |
Funding NO but Skaraborg institute was funding this research. Is this ok?
|
Funding should be YES. Skaraborg institute was funding this research. We are grateful to the reviewer for observing this, our mistake. |
the references must be reviewed. |
We have reviewed the references. |

Reviewer 2 Report
The autors present a well-defined method and in that it contributes to the methodological development within the hermeneutical research in perioperative caring science. This method I believe is appropiate for attaining a deeper understanding of perioperative caring. But some paragraphs could be less extensive. And the conclusions can be improved.
Author Response
Authors revision letter, Manuscript ID: humanities-493715
Thank you for your valuable comments to our manuscript, and for given us the opportunity to review and improve the manuscript. We have considered all recommendations and comments from the reviewers, below you find a table with our point by point responses. The changes made in the manuscript are highlighted in bold font.
From the reviewer 2:
|
Response |
The authors present a well-defined method and in that it contributes to the methodological development within the hermeneutical research in perioperative caring science. |
Thank you |
This method I believe is appropriate for attaining a deeper understanding of perioperative caring. |
Thank you- we agree, that this method Is appropriate for attaining a deeper understanding of perioperative caring. |
But some paragraphs could be less extensive. |
We agree. Several paragraphs have been revised and shortened |
And the conclusions can be improved.
|
We agree. The conclusion has been revised and rewritten in relation to qualities, required by nurses to be in a common world. |

Round 2
Reviewer 1 Report
This is now a well written and interesting paper.
Author Response
Reply to Reviewer 1
Authors revision letter, Manuscript ID: humanities-493715. Thank you for your valuable comments to our manuscript, as now being considered as a well written and interesting paper.